# Probabilistic Orientation Estimation with Matrix Fisher Distributions

**David Mohlin**
KTH/Tobii
Stockholm, Sweden
davmo@kth.se

**Gérald Bianchi**
Tobii
Danderyd, Sweden
gerald.bianchi@tobii.com

**Josephine Sullivan**
KTH
Stockholm, Sweden
sullivan@kth.se

## Abstract

This paper focuses on estimating probability distributions over the set of 3D rotations $(SO(3))$ using deep neural networks. Learning to regress models to the set of rotations is inherently difficult due to differences in topology between $\mathbb{R}^N$ and $SO(3)$. We overcome this issue by using a neural network to output the parameters for a matrix Fisher distribution since these parameters are homeomorphic to $\mathbb{R}^9$. By using a negative log likelihood loss for this distribution we get a loss which is convex with respect to the network outputs. By optimizing this loss we improve state-of-the-art on several challenging applicable datasets, namely Pascal3D+, ModelNet10-$SO(3)$. Our code is available at https://github.com/Davmo049/Public_prob_orientation_estimation_with_matrix_fisher_distributions

## 1 Introduction

Estimating the 3D rotation of an object from 2D images is one of the fundamental problems in computer vision. Several applications relying on 3D rotation estimation have been developed such as a robot grasping an object[23], a self driving vehicle constantly sensing its surrounding environment [18], an augmented reality system combining computer-generated information onto the real world [16], or a system detecting the face orientation to enhance human-computer interactions [25].

Advances of deep learning techniques have resulted in improvements in estimation of 3D orientation. However, precise orientation estimation remains an open problem. The main problem is that the space of all 3D rotations lies on a nonlinear and closed manifold, referred to as the special orthogonal group $SO(3)$. This manifold has a different topology than unconstrained values in $\mathbb{R}^N$, where neural network outputs exist. As a result it is hard to design a loss function which is continuous without disconnected local minima. For example using euler angles as an intermediate step causes problems due to the so-called gimbal lock. Quaternions have a double embedding giving rise to the existence of two disconnected local minimas. Some more complicated methods use Gram-Schmidt [30] which has a continuous inverse, but the function is not continuous with a discontinuity when the input vectors do not span $\mathbb{R}^3$.

Despite these issues various deep learning based solutions have been suggested. One approach is to use one of the rotation representations and model the constraint in the loss function or in the network architecture [14]. An alternative is to construct a mapping function, which directly converts the network output to a rotation matrix [30].

Quantifying the 3D orientation uncertainty when dealing with noisy or otherwise difficult inputs is also an important task. Uncertainty estimation provides valuable information about the quality of the prediction during the process of decision making. Only recent efforts have been made on modeling the uncertainty of 3D rotation estimation [20, 5]. However, those methods still rely on complex solutions to fulfil the constraints required by their parameterization.

In this paper, we instead propose a deep learning approach to estimate the 3D rotation uncertainty by using the matrix Fisher probability density function developed in the field of directional statistics [17]. This unimodal distribution has been selected because of its relevant properties in regards to the problem of 3D orientation estimation: i) The parameterization is unconstrained so there is no need for complex functions to enforce constraints. ii) It is possible to create a loss for this distribution which has desirable properties such as convexity iii) The mode of the distribution can subsequently be estimated along with the uncertainty around that mode for further analysis.

Our method offers a simple solution to the problem of 3D orientation estimation, where a neural network learns to regress the parameters of the matrix Fisher distribution. While several other losses used for rotation estimation are discontinuous with non path connected sublevel sets, with respect to the network output. In this paper we instead propose a loss which is convex with bounded gradient magnitudes, resulting in a stable training. In addition, we also implement a method for computing the the non-trivial normalizing constant of the distribution. Finally, the proposed method is evaluated on multiple problem domains and compared with the latest published approaches. The results show that our method outperforms all previous methods for several problems.

Our contributions include: 1) a method for estimating a probability distributions over the set of rotations with neural networks by using the matrix Fisher distribution, 2) a loss associated with this distribution and show it is convex with bounded gradients, and 3) an extensive analysis encompassing several datasets and recent orientation estimation works, where we demonstrate the superiority of our method over the state-of-the-art.

## 2    Related Work

3D rotation estimation has been studied over the last two decades. A common method estimates 3D rotation by aligning two sets of 3D feature points where each data set is matched and defined in a different coordinate system [4]. Another well-known approach matches 2D keypoints extracted from images with features of a known 3D model and recovers the 3D pose given the 2D-3D correspondences [21, 10]. With the advances of deep learning, the detection of 2D keypoints has significantly been improved [2]. The keypoints can be associated either with physical points [19] or with virtual points such as the corners of an object's bounding box [6].

Recent methods using deep networks often predict 3D rotation directly from images without the knowledge of a 3D model of the object. Those methods can be grouped in two categories. The first one divides the set of rotations into regions and subsequently solves the 3D orientation estimation as a classification task. Subsequently, the classification network output is refined by a regression network [15, 12].

The second category transforms the network output to a 3D rotation representation and learns to directly regress the 3D rotation given an image input. Commonly, quaternions [28] or Euler angles [14] representation are used. However, the paper [30] shows any rotation representation of dimensions four or lower is discontinuous, which makes it difficult for the neural network to generalize over the set of rotations. They propose two continuous 5D and 6D rotation representations and construct a function that maps those representations to a rotation matrix.

Recently some studies have investigated the prediction of rotation uncertainty using probability distributions over rotations. In [20] the parameters of a mixture of von Mises distribution using a biternion network are estimated. In [5], the Bingham distribution over quaternions is used to jointly estimate a probability distribution over all rotation axes. However, their parameters have to be positive semidefinite due to their choice of probability distribution.

In this paper, we propose a solution which learns to regress the probability distribution with unconstrained parameters leading to a simple formulation of the problem of 3D rotation estimation.

## 3    Method

We train a neural network to estimate the 3D orientation of objects in an input image. Specifically, the network outputs the parameters of the matrix Fisher distribution, which is a distribution over $SO(3)$. From the predicted parameters we can obtain the maximum likelihood estimate of the input's orientation. In the rest of this section we review the matrix Fisher distribution and provide

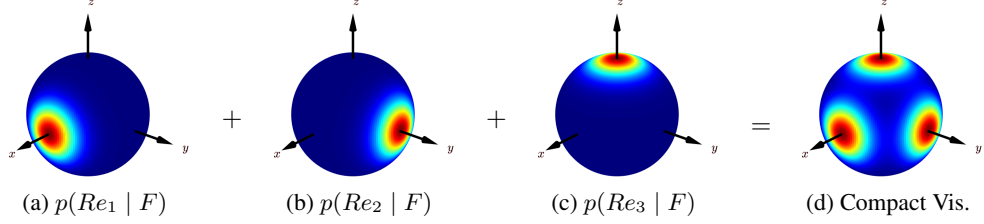

(a) $p(Re_1 \mid F)$     (b) $p(Re_2 \mid F)$     (c) $p(Re_3 \mid F)$     (d) Compact Vis.

Figure 1: **Visualizing the matrix Fisher distribution on** $SO(3)$**.** We follow the convention of [9] and recreate their figures to explain the approach, similarly for figure 2. For the above plots the parameter matrix is $F = \text{diag}(5, 5, 5)$. Let $e_1, e_2$ and $e_3$ correspond to the standard basis of $\mathbb{R}^3$ and is shown by the black axes. (a) This plot shows the probability distribution of $Re_1$ when $R \sim \mathcal{M}(F)$. Thus the pdf shown on the sphere corresponds to the probability of where the $x$-axis will be transformed to after applying $R \sim \mathcal{M}(F)$. (b) and (c) Same comment as (a) except consider $e_2$ and $e_3$ instead of $e_1$. (d) A compact visualization of the plots in (a), (b) and (c) is obtained by summing the three marginal distributions and displaying them on the 3D sphere. All four plots are plotted within the same scale and a *jet* colormap is used.

some visualizations to help the reader's interpretation of its parameters. Then we derive the loss, based on maximizing the likelihood of the labelled data, and finally explain how we deal with the distribution's complex normalizing constant when we calculate our loss and calculate the gradient during back-propagation.

### 3.1 The matrix Fisher distribution on $SO(3)$

We model 3D rotation matrices probabilistically with the matrix Fisher distribution [3, 8]. This distribution has probability density function

$$p(R \mid F) = \frac{1}{a(F)} \exp(\text{tr}(F^T R)) \tag{1}$$

where $F$ is an unconstrained matrix in $\mathbb{R}^{3 \times 3}$ parametrising the distribution, $R \in SO(3)$, and $a(F)$ is the distribution's normalizing constant. We will denote that $R$ is distributed according to a matrix Fisher distribution with $R \sim \mathcal{M}(F)$. The distribution is unimodal but visualizing the distribution in equation (1) is hard as it has a 3D domain. Fortunately, [9] describes a helpful visualization scheme, see figure 1 for details, which we use throughout the paper.

Also not immediately apparent is how the shape of the distribution varies as $F$ varies. From [4] we know the mode of the distribution can be computed from the singular value decomposition of $F = USV^T$, where the singular values are sorted in descending order, and setting

$$\hat{R} = U \begin{bmatrix} 1 & 0 & 0 \\ 0 & 1 & 0 \\ 0 & 0 & \det(U\,V) \end{bmatrix} V^T \tag{2}$$

This is done to ensure that the orientation $\hat{R}$ has determinant 1 and is orthonormal. Similar results are available in [3]. Figure 2 displays examples of the distribution for simple $F$ matrices. These figures show that larger singular values correspond to more peaked distributions. To further help understanding of how the shape of the distribution relates to $F$ please consult section 3 of the supplementary material and [9].

Finally, the normalizing function $a(F)$ in equation (1) can be defined as in equation (3). This function, as well as its gradients, can be computed by doing an integral over Bessel functions[9]. For implementation details regarding this see supplementary material section 5.

$$a(F) = \int_{R \in SO(3)} \exp(\text{tr}(F^T R)) \, dR \tag{3}$$

z

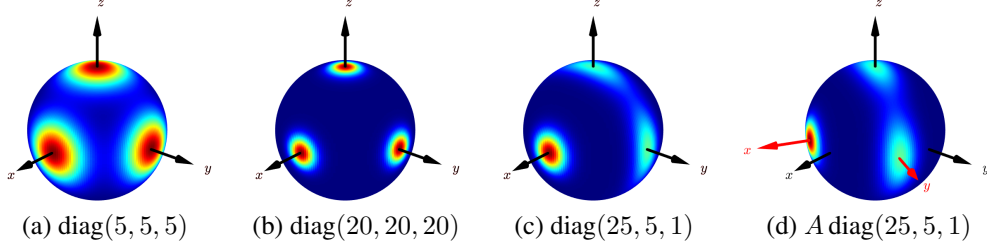

| (a) $\mathrm{diag}(5,5,5)$ | (b) $\mathrm{diag}(20,20,20)$ | (c) $\mathrm{diag}(25,5,1)$ | (d) $A\,\mathrm{diag}(25,5,1)$ |

Figure 2: **Visualization of the matrix Fisher distribution for simple $F$ matrices**. (a) For a spherical $F$ the mode of the distribution is the identity. The distribution for each axis is circular and identical. (b) Here the axis distributions are more peaked than in (a) as the singular values are larger. (c) The distribution for the $y$- and $z$-axes are more elongated than for the $x$-axis as the first singular value dominates. (d) $A$ is the rotation matrix obtained by rotating around the $z$-axis by $-\pi/6$ degrees and thus the mode rotation is $A$ shown by the red axes. The shape of the axes distributions though remains as in (c).

## 3.2  A negative log-likelihood loss function

Assume we have a labelled training example $(x, R_x)$ where $x$ is the input and $R_x \in SO(3)$ its ground truth 3D rotation matrix. To train a neural network that estimates $F_x$ for input $x$, it is necessary to define a loss function measuring the compatibility between $F_x$ and $R_x$. As the pdf in equation (1) has support in all of $SO(3)$, we use the negative log-likelihood of $R_x$ given $F_x$ as the loss:

$$\mathcal{L}(F_x, R_x) = -\log(p(R_x \mid F_x)) = \log(a(F_x)) - \mathrm{tr}(F_x^T R_x) \qquad (4)$$

This loss has several interesting properties such as it is Lipschitz continuous, convex and has Lipschitz continuous gradients which makes it suitable for optimization. See supplementary material section 4 for proofs.

In practice the loss in equation 4 has an equilibrium far from the origin, in some experiments we believe this led to instability. To alleviate this problem we used a regularizing term which was 2.5% larger than what is analytically correct to move the equilibrium closer to the origin.

## 4  Experimental Details

We test our proposed approach on three separate datasets Pascal3D+, ModelNet10-$SO(3)$ and UPNA head pose. We briefly describe these datasets, the pre-processing we applied to the images from each dataset before training and then the evaluation metrics.

### 4.1  Datasets & Pre-processing

**Pascal3D+** [27] has 12 rigid object classes and contains images from Pascal VOC and ImageNet of these classes. Each image is annotated with the object's class, bounding box and 3D pose. The latter is found by having an annotator align a 3D CAD model to the object in the image.

We pre-process each image by applying a homography so that the transformed image appears to come from a camera with known intrinsics and a principal axis pointing towards the object. This approach is similar to [13] and [29]. More details are given in section 6 of the supplementary material. We perform data augmentation similar to the data augmentations introduced in [14], but adapted for our preprocessing. At test time we apply the same type of homography transformation as applied during training, but no data augmentation.

**ModelNet10-$SO(3)$**[12] is a synthetic dataset. It is created by rendering rotated 3D models from ModelNet10 [26] with uniformly sampled rotations. The task is to estimate the applied rotation matrix.

We do not use any preprocessing for these images since the object is already centered and of a reasonable size. We do not use any data augmentation as the original paper did not use any and we want a fair comparison between the losses.

**UPNA head pose** [1] consists of videos with synchronized annotations of keypoints for the face in the image as well as its 3D rotation and position. The dataset has 10 people each with 11 recordings.

From the keypoint annotations we create a face bounding box for each image. After this we perform a small random perturbation of this bounding box to degrade the quality of the bounding box to be similar to what one would expect to get from a face detector. Using this artificial bounding box enables us to use the same data augmentation and preprocessing as we used for Pascal3D+.

There is no official training/test split for this dataset. We use a test split with the same people held out as in prior work [5]. We did not use a validation set since we did not do a new hyperparameter search for the dataset.

## 4.2 Details of network & training

We run experiments with ResNet-101 as our backbone network. The ResNet-101 parameters are initialized from pre-trained ImageNet weights. The object's class is encoded by an embedding layer that produces a 32-dimensional vector and which is appended to the ResNet's activations obtained from the final average pooling layer. We apply 3 fully connected layers to this vector with $[512, 512, 9]$ nodes output at each layer. We use pytorch's implementation of SVD for forward and backward propagation.

We fine-tune the embedding and fully connected layer weights for 2 epochs. We use SGD and start with a learning rate of 0.01. We use a batch size of 32 and train for 120 epochs. For Pascal3D+ we reduce this learning rate by a factor 10 at epochs 30, 60 and 90. For ModelNet10-SO(3) we train for 50 epochs and reduce the learning rate by a factor of 10 at epochs 30, 40 and 45. For UPNA head pose we use the same hyperparameters as for Pascal3D+, except we do not use a class embedding since there are only faces in this dataset.

## 4.3 Evaluation metrics

The evaluation metrics used are based on the geodesic distance:

$$d(R, \hat{R}) = \arccos\left(\frac{1}{2}\left(\text{tr}(R^T\hat{R}) - 1\right)\right) \tag{5}$$

where $R$ and $\hat{R}$ are the ground truth and estimated rotation respectively. This metric returns an angle error and we measure it in degrees. For a test set $\mathcal{X}$, containing tuples $(x, R_x)$ of input $x$ and its ground truth rotation $R_x$, we summarize performance on $\mathcal{X}$ with the median angle error and Acc@$Y$:

$$\text{Acc@}Y = \frac{1}{|\mathcal{X}|} \sum_{(x, R_x) \in \mathcal{X}} \mathbb{1}(d(R_x, \hat{R}_x) < Y) \tag{6}$$

where $\mathbb{1}(\cdot)$ is the indicator function, $\hat{R}_x$ is the estimated rotation for input $x$ and $|.|$ is the cardinality of a set. To compute the overall performance on a dataset the median angle error and Acc@$Y$ are first computed per class and then averaged across all classes. For the UPNA dataset we use the mean geodesic error angle instead of the median to allow more direct comparison with the results in [5].

We have done all development and hyper-parameter optimization where the full training set was partitioned into a training and validation set. After hyper-parameter optimization, we have used the full training set for training and evaluated on the test set to get the numbers presented in the tables. For the Pascal3D+ dataset, we use the ImageNet validation split for the test set. Some samples of Pascal3D+ are labeled as "truncated", "difficult" or "occluded". We exclude these samples from our evaluations similar to other reported results [12]. This implementation detail had only a very slight effect on performance.

## 5 Results

**Quantitative results**   Table 1 compares the performance of our method and recent high performing approaches. Table 2 compares per class performance for some classes on Pascal3D+ with the previous state-of-the-art method [15]. Our method significantly outperforms all the prior approaches. When

Table 1: **Performance on Pascal3D+**. Results are reported for the median angle error, Acc@$\pi/6$ and Acc@$\pi/12$. The last column indicates if the training set was augmented with the synthetic dataset from [22].

| Method | MedErr | Acc@$\pi/6$ (%) | Acc@$\pi/12$ (%) | Use synth. |
|---|---|---|---|---|
| Mahendran et al. [14] | 15.38 | – | – | × |
| Pavlakos et al. [19] | 14.16 | – | – | × |
| Tulsiani and Malik [24] | 13.60 | 81.0 | – | × |
| Su et al. [22] | 11.70 | 82.0 | – | ✓ |
| Grabner et al. [6] | 10.90 | 83.9 | – | × |
| Prokudin et al. [20] | 10.40 | 83.9 | – | × |
| Mahendran et al. [15] | 10.10 | 85.9 | – | ✓ |
| Liao et al. [12] | 9.20 | 88.7 | – | ✓ |
| Ours | 9.11 | 90.9 | 73.4 | × |
| Ours | **8.17** | **92.8** | **77.8** | ✓ |

the training set is augmented with the synthetic dataset from [22], we further reduce the mean over medians angle error by approximately 1 degree.

The results reported in table 3 show that our method also achieves state-of-the-art performance on ModelNet10-SO(3).

On the UPNA head pose dataset our algorithm gives a mean angle error of 6.5 degrees. This is on par with current state of the art who quote a performance of 6.3 degrees [5]. We do not think the differences between these two methods are significant since there are only 4 persons in the test set.

**Qualitative results**  Figure 3 displays and discusses interesting qualitative results on Pascal3D+ which highlight the probabilistic performance of our method.

**Behaviour for classes with rotational symmetries**  Several classes in the datasets used have rotational symmetries or effectively have rotational symmetries due to very similar appearance at several distinct viewpoints. Some examples of these classes are canoes, bathtubs, tables, desks, and bottles. Our modelling though is based on a unimodal distribution and here we describe how the model copes with the inherent ambiguity of rotational symmetric objects.

For Pascal3D+ our method performs well, somewhat surprisingly, for the *dining table* class, see table 2. However, when the synthetic data is used to augment the training data the performance on this class drops. We suspect the manual labeling process introduces biases for this class with one of the ambiguous poses being labelled much more frequently. But the synthetic data added does not have these biases. This discrepancy between the distribution of training and test set label results in the drop in performance. For ModelNet10-SO(3) the *table* and *bathtub* classes have rotational symmetries and thus these two classes have much higher median errors than the other classes, see table 4. In figure

Table 2: **Pascal3D+ per-class performance** of our method, with or without using extra synthetic training data, compared to the competitive method Mahendran et al. [15]. The top three rows report the median angle error per class measured in degrees. The bottom three rows report Acc@$\pi/6$ measured as a percentage.

| Method | aero | bike | boat | bottle | bus | chair | dtable | sofa | train | **mean** |
|---|---|---|---|---|---|---|---|---|---|---|
| [15] | 8.5 | 14.8 | 20.5 | **7.0** | **3.1** | 9.3 | 11.3 | 10.2 | 5.6 | 10.1 |
| Ours w/o | 10.1 | 14.6 | 13.2 | 8.0 | 3.3 | 7.4 | **8.2** | 8.2 | 5.8 | 9.1 |
| Ours with | **6.6** | **12.5** | **11.6** | 7.7 | 3.5 | **6.6** | 11.2 | **7.4** | **5.3** | **8.2** |
| [15] | 87.0 | 81.0 | 64.0 | **96.0** | 97.0 | 92.0 | 67.0 | 97.0 | 82.0 | 85.9 |
| Ours w/o | 87.7 | 83.2 | 75.6 | 94.9 | 98.6 | 93.9 | **82.3** | 97.4 | 97.9 | 87.7 |
| Ours with | **92.9** | **88.5** | **80.7** | 95.1 | **99.0** | **98.7** | 76.5 | **99.0** | **98.0** | **92.8** |

Table 3: **Performance on ModelNet10-SO(3)**. * indicates the numbers reported in the original paper, but † denotes the revised numbers [11] where the evaluation metric uses the distance defined in equation (5). Thus we compare the performance of our method to the latter numbers.

| Method | MedErr (deg) | **Acc@$\pi/6$ (%)** | Acc@$\pi/12$ (%) | Acc@$\pi/24$ (%) |
|---|---|---|---|---|
| Liao et al. [12]* | 20.3 | 70.9 | 58.9 | 38.4 |
| Liao [11]† | 28.7 | 65.8 | 49.6 | 35.2 |
| Ours | **18.0** | **75.2** | **68.5** | **53.9** |

Table 4: **Per class performance on ModelNet10-SO(3)** of our method.

| **Metric** | bathtub | bed | chair | desk | dress. | t.v. | n. stand | sofa | table | toilet |
|---|---|---|---|---|---|---|---|---|---|---|
| MedErr | 86.5 | 4.4 | 5.2 | 13.7 | 6.9 | 6.1 | 15.4 | 4.1 | 34.3 | 3.9 |
| Acc@$\pi/6$ | 41.1 | 90.0 | 93.7 | 67.7 | 72.8 | 85.8 | 59.1 | 94.8 | 49.2 | 98.2 |
| Acc@$\pi/12$ | 32.2 | 87.0 | 88.7 | 52.9 | 65.9 | 78.4 | 49.5 | 91.7 | 43.4 | 95.8 |

4(f) the histogram of angle errors for the *table* class has a "U" shape and the median is in the middle of this "U". This histogram indicates that at test time the network predicts one of the relevant poses.

To further illuminate this point, plots (b)-(g) of figure 4 show the evolution of the distribution predicted during training for one specific *table* class test image. The axis which has no associated ambiguity is identified correctly and confidently early on in training. The other two directions are predicted to have an almost uniform distribution on the plane spanned by the ambiguous axes. This is arguably the best way for the unimodal distribution to describe the situation. In the latter stages of training the network correctly identifies the object's full pose on the training set and uncertainty becomes small. Such behaviour should be considered as a deterioration of the network's probabilistic modelling as it effectively randomly chose one pose from the set of plausible poses and report it is very confident about this decision. Continuing to improve the accuracy on the test set while overfitting the loss often occurs with cross-entropy training of classification networks as well [7]. The dataset's accuracy and loss plots, in the supplementary material section 2, show our loss is susceptible to this trend too.

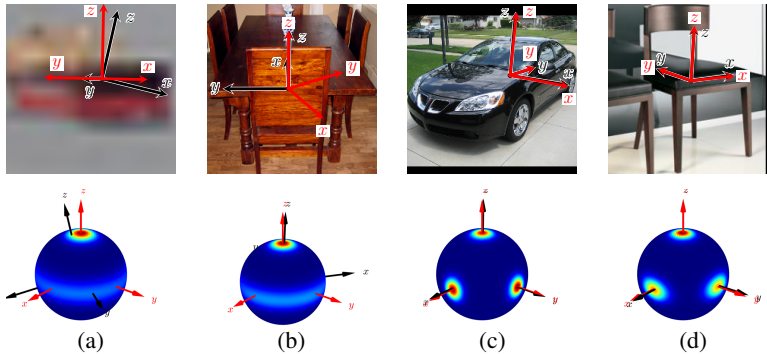

(a)  (b)  (c)  (d)

Figure 3: **Interesting qualitative results on Pascal3D+**. The top row displays example input images with the projected axes displaying the predicted pose (red) and labelled pose (black) of the object. The bottom row shows a visualization of the pdf estimated by the network. The red axis show the maximum likelihood estimate of the rotation matrix estimated from the predicted $F$ matrix, while the axis in black corresponds to the ground truth rotation/pose. For clarity we have aligned the predicted pose with the standard axis. Each probability plot has been scaled independently. The examples shown have been specifically chosen to highlight our algorithm's performance for certain cases: (a)-(b) Examples where model has high uncertainty for the azimuth either due to low resolution or rotation symmetry (c)-(d) Examples where model predicts rotations with high certainty and reasonably low errors.

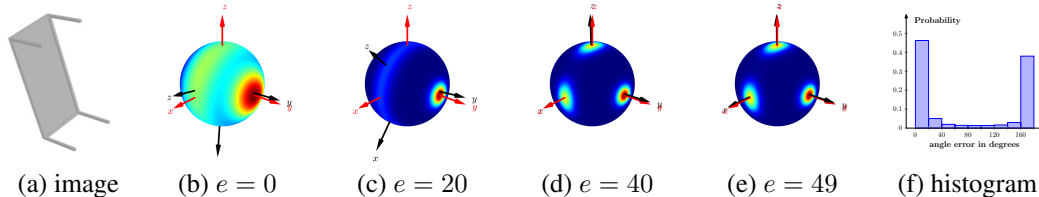

|   (a) image   |   (b) $e = 0$   |   (c) $e = 20$   |   (d) $e = 40$   |   (e) $e = 49$   |   (f) histogram   |

Figure 4: **Evolution of the estimated pdf during training for a rotational symmetric object**. The leftmost figure is a test image from the table class. (b)-(e) Each plot displays the predicted distributions for the object's pose after $e$ epochs of training. The mode of the distribution is shown in red and the ground truth rotation in black. (f) Histogram of the test error angle for the table class after 50 epochs. See the main text for comments.

**Ablation experiments**    We run ablation experiments on Pascal3D+ to identify the importance of the individual components of our approach. The factors considered are data-augmentation, the class embedding and pre-processing the image via a homography. The results are shown in table 5. Row one in table 5 shows the performance for a very simple method which uses a standard network architecture and plain cropping without data augmentation as preprocessing. This method gets higher performance than several more complicated methods, See table 1. This indicates that the loss we have introduced is a significant improvement by itself.

By comparing Row one and two in table 5 we see that the class embedding does not seem to give any improvement for this dataset. This can also be seen by comparing row four and five. Rows one and three in table 5 show that our warping does not provide a significant improvement by itself on Pascal3D+ compared to cropping. We believe though this warping could be advantageous in many situations and therefore should be used irrespective of these results. In theory this pre-processing should allow our method to generalize across all pinhole cameras with known intrinsic parameters and negligible radial distortion rather than for cameras with the same intrinsics as Pascal3D+.

Comparing row three with row four or comparing row one with row five show that data augmentation gives a significant improvement. This is consistent with prior work.

Table 5: **Results of ablation experiments on Pascal3D+** for our method.

| Data aug. | Class embed | Crop | Warp | MedErr (deg) | Acc@$\pi/6$ (%) | Acc@$\pi/12$ (%) |
|:---:|:---:|:---:|:---:|:---:|:---:|:---:|
| × | × | ✓ | × | 10.5 | 87.7 | 68.6 |
| × | ✓ | ✓ | × | 10.5 | 87.1 | 67.9 |
| × | ✓ | × | ✓ | 10.5 | 87.0 | 68.8 |
| ✓ | ✓ | × | ✓ | 9.2 | **90.9** | 73.4 |
| ✓ | × | × | ✓ | **9.0** | 90.5 | **74.1** |

## 6   Conclusion & Future work

In this paper we have introduced a way to use neural networks to output probability distributions over the set of rotations with the matrix Fisher distribution. We show when optimizing the negative log likelihood of this distribution we end up with a convex loss. When applying this method on several datasets we get state-of-the-art performance. Our ablation studies show the relative robustness of the approach.

Since the matrix Fisher distribution is unimodal it poorly models classes which have rotational symmetries. It could be interesting to try to create a loss supporting multimodal distributions while keeping the desirable optimization properties of our loss. It could be possible to use these estimated probabilities for time tracking filters such as the one described in [9].

We have not done any quantification of how well the estimated variances correspond to the actual errors. Doing this as well as calibrating the uncertainties similar to [7] is potential future work.

## Broader Impact

The methods described in this paper has obvious applications in fields which some consider ethically questionable such as for surveillance and military systems. One example could be determining heading for ships or airplanes for tactical planning. That being said, the orientation of objects is a fundamental property of objects in the real world and being able to accurately estimate this property should be helpful for many applications of either an ethically desirable or undesirable nature. In our opinion improving the techniques used for orientation estimation has a similar societal impact as improving the techniques used for classification or object detection.

The persons in the UPNA dataset are unlikely to be sampled from a uniform distribution of people across the world, for this reason one can not expect the reported performance to be accurate for the world population in general, that being said due to the small test size this reported performance might not reflect the average performance for any population. We do not believe this is an issue since models for predicting head pose which are deployed on a wider scale are very unlikely to use this dataset due to its small size and non-commercial licence. The method itself is not reliant on any population specific feature.

## Acknowledgments and Disclosure of Funding

DM is an industrial PhD student at Tobii AB and GB is a senior algorithm engineer at Tobii AB. This work was partially supported by the Wallenberg AI, Autonomous Systems and Software Program (WASP) funded by the Knut and Alice Wallenberg Foundation. Computational resources was provided by KTH.

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
