[Supplementary Material]

Table 1: **Pascal3D+ per-class performance** of our method, w/o and with synthetic training data, compared to the state-of-the-art method Mahendran et al. [5] on this dataset. The top three rows report the median angle error per class measured in degrees. The bottom three rows report Acc@$\pi/6$ measured as a percentage.

| Method | aero | bike | boat | bottle | bus | car | chair | dtable | mbike | sofa | train | tv | **mean** |
|---|---|---|---|---|---|---|---|---|---|---|---|---|---|
| [5] | 8.5 | 14.8 | 20.5 | **7.0** | **3.1** | 5.1 | 9.3 | 11.3 | 14.2 | 10.2 | 5.6 | 11.7 | 10.1 |
| Ours w/o | 10.1 | 14.6 | 13.2 | 8.0 | 3.3 | 4.0 | 7.4 | **8.2** | 14.6 | 8.2 | 5.8 | 11.6 | 9.1 |
| Ours with | **6.6** | **12.5** | **11.6** | 7.7 | 3.5 | **3.9** | **6.6** | 11.2 | **10.4** | **7.4** | **5.3** | **11.4** | **8.2** |
| [5] | 87.0 | 81.0 | 64.0 | **96.0** | 97.0 | 95.0 | 92.0 | 67.0 | 85.0 | 97.0 | 82.0 | 88.0 | 85.9 |
| Ours w/o | 87.7 | 83.3 | 75.6 | 94.9 | 98.6 | 98.3 | 93.9 | **82.3** | 92.4 | 97.4 | 97.9 | 88.5 | 90.9 |
| Ours with | **92.9** | **88.5** | **80.7** | 95.1 | **99.0** | **99.0** | **98.7** | 76.5 | **93.7** | **99.0** | **98.0** | **92.8** | **92.8** |

# 1 Additional experimental results

## 1.1 Additional quantitative results

In table 1 we show the quantitative performance for our method across all classes on the Pascal3D+ test set.

## 1.2 Additional qualitative results

Figure 1 shows how well our method works on a random subset of the Pascal3D+ test set.

Figure 1: **Pascal3D+ predictions.** Visualization of a random subset of the test set for Pascal3D+. The maximum likelihood rotation for the probability distribution predicted by the network (thick lines) compared to the ground truth (thin lines). Background image is the preprocessed input

Figure 2: **Median error compared to mean loss** Visualization of how the median error and median loss changes over time when training on ModelNet10-SO(3)

## 2 Overfitting confidences

In figure 2 we see how we overfit more on the loss than the median error. A similar effect have been observed when training networks to perform classification [1]. This effect was more pronounced for ModelNet10-SO(3) due to the prominence of rotation ambiguous samples.

## 3 Geometric interpretation of the matrix Fisher distribution

To further help understanding how the shape of the distribution relates to $F$, [4] defines the *proper SVD* of $F$ as:

$$F = U_1 S' V_1^T = U_1 \underbrace{\begin{bmatrix} 1 & 0 & 0 \\ 0 & 1 & 0 \\ 0 & 0 & \det(U_1) \end{bmatrix}}_{U} \underbrace{\begin{bmatrix} s_1' & 0 & 0 \\ 0 & s_2' & 0 \\ 0 & 0 & \det(U_1 V_1)s_3' \end{bmatrix}}_{S} \underbrace{\begin{bmatrix} 1 & 0 & 0 \\ 0 & 1 & 0 \\ 0 & 0 & \det(V_1) \end{bmatrix}}_{V^T} V_1^T = USV^T \quad (1)$$

so that $U$ and $V$ are guaranteed to be rotation matrices and $S$ contains the *proper singular values* of $F$ with $s_1 \geq s_2 \geq |s_3|$. Note that $\hat{R} = UV^T$. The columns of $U$ define three orthogonal axes directions around which the mode rotation can be rotated. When rotating $\hat{R}$ around the axis $Ue_i$ the peakedness of the pdf is given by $(s_j + s_k)$ with $j, k \in \{1, 2, 3\}\backslash i$. As $(s_j + s_k)$ approaches zero, then the distribution approaches a uniform distribution with respect to this rotation. On the other hand if $(s_j + s_k)$ increases towards $\infty$ then this distribution approaches a Dirac function with respect to this rotation. The columns of $U$ are termed the *principal axes* and can be thought of as analogous to the *principal axes* of a multivariate Gaussian. Figure 4 gives a visualization of examples of this interpretation.

(a) $p(Re_1 \mid F)$      (b) $p(Re_2 \mid F)$      (c) $p(Re_3 \mid F)$      (d) Compact Vis.

Figure 3: **Visualizing the matrix Fisher distribution on** $SO(3)$**.** Visualization of the probabilities for the case $F = \text{diag}(5, 5, 5)$. Let $e_1, e_2$ and $e_3$ correspond to the standard basis of $\mathbb{R}^3$ and is shown by the black axes. (a) This plot shows the probability distribution of $Re_1$ when $R \sim \mathcal{M}(F)$. Thus the pdf shown on the sphere corresponds to the probability of where the $x$-axis will be transformed to after applying $R \sim \mathcal{M}(F)$. (b) and (c) Same comment as (a) except consider $e_2$ and $e_3$ instead of $e_1$. (d) A compact visualization of the plots in (a), (b) and (c) is obtained by summing the three marginal distributions and displaying them on the 3D sphere. All four plots are plotted within the same scale and a *jet* colormap is used.

(a) $\text{diag}(20, 20, 20)$      (b) $\text{diag}(25, 5, 1)$      (c) $A_3 \text{diag}(25, 5, 1)$

(d) $A_1 \text{diag}(25, 5, 1) A_1^T$    (e) $A_2 \text{diag}(25, 5, 1) A_2^T$    (f) $A_3 \text{diag}(25, 5, 1) A_3^T$

Figure 4: **Effect of $F$ on shape of the matrix Fisher distribution**. Below each plot is the value of $F$. Each $A_i$ corresponds to the rotation matrix obtained by rotating by $-\pi/6$ degrees around $e_i$. (a) For a spherical $F$ the mode of the distribution is the identity and the principal axes correspond to $e_1, e_2, e_3$. The distribution for each axis is circular and identical and more peaked than in figure 3 as the singular values are larger. (b) Have a diagonal $F$ and thus the mode and the principal axes once again coincide. The distributions for the $y$- and $z$-axes are more elongated than for the $x$-axis as the first singular value dominates. (c) Here $F$ is obtained by pre-multiplying a diagonal matrix by $A_3$. Thus the mode corresponds to $A_3$ and is shown by the red axes. The shape of the distributions for each axis though remain the same as in (b). (d, e, f) In each of these plots the diagonal matrix from (b) is pre-multiplied by $A_i$ and post-multiplied by $A_i^T$. Thus the mode rotation is the identity. However, the principal axes correspond to the columns of $A_i$. Thus the axis distributions are centred at the standard location but the orientation of the spread has been affected by the direction of the principal axes.

# 4 Properties of loss

For more convenient notation we will introduce a flattening function $h : \mathbb{R}^{m \times m} \to \mathbb{R}^{m^2}$ s.t. $h(x)_{(i-1)*m+j} = x_{i,j} \forall i, j \in \{1, 2, \cdots m\}$ and an inflation function $g$ such that $g = h^{-1}$.

In this section F is an element of $\mathbb{R}^9$. $||.||_F$ is the frobenius norm. $||.||_2$ for vectors is the traditional $L_2$ norm. At one place we use the matrix 2 norm which is the magnitude of the largest eigenvalue. We will use the nonstandard notation $||.||_{M2}$ for this norm to avoid confusion. The standard notation for this norm is $||.||_2$. We will use the fact that $tr(g(F)^T R) = F^T h(R)$ and $||g(F)||_F = ||F||_2$

## 4.1 Lipschitz continous

Here we show that the loss is $\alpha$-Lipschitz continous for $\alpha$=6. This is equivalent to the $L_2$ norm of the gradient being less than 6.

$$Loss(F, R) = log(a(g(F))) - tr(g(F)^T R) \tag{2}$$

The gradient is

$$||\nabla_F Loss(F, R)||_2 = ||\nabla_F log(a(g(F))) - h(R)||_2 \leq ||\nabla_F log(a(g(F)))||_2 + || - h(R)||_2 \tag{3}$$

The last step follows from triangle inequality

We know $||h(R)||_2 = ||R||_F = 3$ since it is a rotation matrix.

We now use the definition of a(F) to compute the gradient

$$\nabla_F log(a(g(F))) = \frac{\nabla_F a(g(F))}{a(g(F))} = \frac{1}{a(g(F))} \nabla_F \int_{h(R) \in SO(3)} \exp(tr(g(F)^T R)) dR = \tag{4}$$

$$\frac{1}{a(g(F))} \int_{R \in SO(3)} h(R) \exp(F^T h(R))) dR = E[h(R)|F] \tag{5}$$

Due to convexity of frobenius norm and Jensen's inequality we have $||\nabla_F log(a(g(F)))||_2 = ||E[h(R)|F])||_2 = ||E[R|F]||_F \leq E[||R||_F|F] = 3$ This concludes the proof.

## 4.2 Convexity

We first compute the hessian of $log(a(g(F)))$

We already have $\nabla_F log(a(g(F))) = \frac{1}{a(g(F))} \int_{R \in SO(3)} h(R) \exp(tr(g(F)^T R)) dR$ from previous section.

We differentiate again to get

$$(\nabla^2 log(a(g(F))))_{i,j} = \frac{1}{a(g(F))} \int_{R \in SO(3)} h(R)_i h(R)_j \exp(tr(g(F)^T R)) dR - \tag{6}$$

$$\frac{1}{(a(g(F)))^2} \int_{R \in SO(3)} h(R)_i \exp(tr(g(F)^T R)) dR \int_{R \in SO(3)} h(R)_j \exp(tr(g(F)^T R)) dR = \tag{7}$$

$$E[h(R)h(R)^T]_{i,j} - (E[h(R)]E[h(R)]^T)_{i,j} = Var[h(R)]_{i,j} \tag{8}$$

Since a variance matrix is positive semidefinite it follows that the hessian is as well. Therefore this term is convex The term $tr(g(F)^T R)$ is linear. Linear functions are convex. The set of convex functions are closed under addition. Therefore the loss is convex with respect to the network output $F$.

## 4.3 Lipschitz continous gradients

A function has $\beta$-Lipschitz continous gradients if the largest eigenvalue of the hessian is less than $\beta$.

We know

$$||\nabla^2 log(a(g(F)))||_{M2} = ||Var[h(R)]||_{M2} \leq ||Var[h(R)]||_F = \tag{9}$$

$$||E[h(R)h(R)^T] - E[h(R)]E[h(R)]^T||_F \leq ||E[h(R)h(R)^T]||_F \leq \tag{10}$$

$$E[||h(R)h(R)^T||_F] \leq E[||h(R)||_F^2] = 9 \tag{11}$$

Therefore this function has $\beta$-Lipschitz continous gradients with $\beta$=9

# 5 Approximating the normalizing constant

The normalizing constant can be expressed as a generalized hypergeometric function of matrix arguments. This can be defined recursively by integrals over positive definite matrices [2]. Similar to the standard generalized hypergeometric function it has a combinatorial definition as well which is

$$_1F_1^{(2)}(\tfrac{1}{2}, 2, X) \sum_{k=0}^{\infty} \sum_{\kappa \vdash k} \frac{(\tfrac{1}{2})_\kappa^{(2)}}{k!(2)_\kappa^{(2)}} C_\kappa^{(2)}(X) \tag{12}$$

For details see [3]. Another way to compute this function is given in [6].

The normalizing constant can also be expressed as a one dimensional integral over Bessel functions as described by the equation (14) and (15) in [4]. We approximate this integral by using the trapezoid rule. In the approximation we use for experiments 511 trapezoids. We use standard polynomials to approximate the Bessel function using Horner's method. Trapezoid integrals and parallel evaluations of Horner's method are simple to implement in a vectorized manner using for example numpy or pytorch, in the latter case to potentially run on a GPU. Our implementation of this approximation has a negligible computational cost compared to the forward and backward pass of a neural network.

To ensure correctness we have checked that the analytical and numerical gradients of the functions are similar, we have also compared this implementation with Koev's implementation [3] to check that the two implementations are consistent.

To approximate the correctness we have used the same implementation (i.e. trapezoid integrals of bessel functions) with $2^{14} - 1$ trapezoids in float128 precision in place of the "true function". The only source of errors which we know remains is the approximation of the bessel function, but here we use a standard method which should have a very high accuracy.

We have evaluated our function on 1000 randomly sampled points with singular values less than 50, with a 50% chance of setting the smallest eigenvalue to be negative. We believe this should cover the values we will encounter during training. We do not think there are any issues with using this method for larger singular values either.

For these experiments we evaluated the accuracy of our forward pass by evaluating $|log(a(S)) - f(S)|$ where $f$ is our approximation of $log(a(S))$. We evaluate the accuracy of the backward pass by evaluating. $||\nabla_S log(a(S)) - g(S)||_2$ where $g$ is our approximation of $\nabla_S log(a(S))$ and $||.||_2$ is the vector 2 norm.

The maximum error encountered in the forward pass was $4.6 * 10^{-3}$. The mean error of the forward pass was less than $1.2 * 10^{-3}$. The maximum error for the backward pass was less than $3.4 * 10^{-3}$. The mean error for the backward pass was less than $6.9 * 10^{-4}$.

# 6 Pascal3D preprocessing

Figure 5: illustration of how naive cropping changes perceived orientation. Left: original image with bounding box. Middle: Cropped image with unchanged ground truth orientation. Right: warped image with adjusted ground truth orientation. As we can see the middle image appears to have a different azimuth due to aspect ratio of bounding box. On the right image the green axis is well aligned with the "backwards" direction of the car.

The flaws of using cropping as a method are twofold. Firstly if the width and height of the bounding box are not the same the scaling can cause artifacts which are very similar to a rotation. For reference see figure 5. Secondly as the object moves away from the principal axis of the camera the cropped image will change in a similar manner compared to when the object is rotated.

To solve these issues we assume that the position of the bounding box of the object is known. We now create a desired pinhole camera which is rotated relative to the real camera in such a way that the principal axis is facing the center of the bounding box. We let the intrinsic of the desired camera be

$$I_{ideal} = \begin{bmatrix} f & 0 & s/2 \\ 0 & f & s/2 \\ 0 & 0 & 1 \end{bmatrix}$$

Where s is the size of the pictures taken with this camera and f is picked to be the largest value such that all points of the bounding box is still inside of the pictures taken with the virtual camera.

The transformation between the desired camera and the real camera is now a homography, we can simulate taking pictures with the desired camera warping the image from the real camera.

When we estimate orientations we estimate them relative to the new camera. This is not cheating since if one wanted the orientation in the camera coordinate system on could just apply the (known) inverse rotation between the two cameras, and the loss and evaluation metric are both invariant to what coordinate system is used.

## 6.1 Camera details

To compute rotation for the desired camera we first backproject every corner of the bounding box onto a sphere by

$$\hat{p} = I_{real}^{-1} \begin{bmatrix} p_y \\ p_x \\ 1 \end{bmatrix} \tag{13}$$

followed by

$$p = \frac{\hat{p}}{||\hat{p}||_2} \tag{14}$$

where $p_x$, $p_y$ is the position in the image and $p$ is the backprojected point on a sphere.

We now have 4 points on a sphere, to find the desired direction of the principal point we apply a modified version of Welzl's algorithm to find the minimal enclosing sphere of these points, subject to the constraint that the center of the enclosing sphere have a center at distance one from the origin.

If the bounding box spans more than 180 degrees one could get a solution from Welzl's algorithm which is pointing 180 degrees in the wrong direction. Since the datasets use normal cameras this will not happen. In addition to this, if the bounding box spans more than 180 degrees it is not possible for a pinhole camera to capture the whole object, due to limitations of the pinhole model.

There is one more degree of freedom, for the rotation of the ideal camera, we eliminate this degree by adding a constraint on the direction for the y axis in the desired image.

We can now choose the focal length to be the largest value constrained by the fact that all bounding box coordinates has to be projected in $[0, s] \times [0, s]$, i.e. visible in the warped image.