[Reviews · NeurIPS 2020]

Review 1

Summary and Contributions: The paper focuses on learning 3D rotations on the SO(3) space in the form of probability distributions. As the authors explain, the SO(3) manifold is non-linear and closed (bounded), whereas deep neural networks normally return unbounded activations. Thus, it is not straightforward how to map neural network activations on the SO(3) manifold without restoring to more explicitly structure in the neural network definition. To this end, the paper proposed to model SO(3) rotations with the Fisher distribution. Although the Fisher distribution is unimodal, meaning that it will not be able to disambiguate between symmetries, it can model general rotations in a flexible manner.

Strengths: - The problem that the paper addresses is very relevant and emerging. While most works so far approached the learning of rotations in a straightforward way, ignoring the geometric properties of the underlying rotation manifolds, this paper explains very clearly and eloquently the necessity for better geometrical representations. - The matrix Fisher distribution is a very good fit to the problem definition of modelling SO(3) rotations, as its parameters have no constraints (similar to what neural networks can model) and the conditioning variable, F, can obtain the parameterizing structure of rotations (R^3x3 matrix). The resulting negative log-likelihood has interesting properties, like being Lipschitz continuous, although unfortunately the paper does not dwell much on this. - Results look consistently better compared to recent state-of-the-art baselines [13, 16] and over a different number of classes.

Weaknesses: - As also the authors explain, the proposed solution for the intractable normalizing constant is inelegant and also rather unclear. It is not clear how the function is approximated, given that the more complex version is not possible to know. Is some variational approximation used to make sure that the obtained function minimizes some distributional metric on the true normalizing constant? Also, is there a way to get an estimate of the crudeness of the approximation, perhaps on a simpler toy setting where one can solve the problem computationally? - Importantly, given that the approximation of the normalizing constant is it still fair to claim that the output predictions lie on the SO(3) manifold? Given the high dimensionality (9 dimensions are still quite high geometrically), is it fair to say that the approximation may lie quite far from the original SO(3) manifold? In that case, what exactly is the manifold learned and how does it relate to the SO(3) manifold? - Given the unimodality of the distribution, what is the convergence behavior of the algorithm? Is it observed that it may get stuck to bad local optima, especially for classes for which there is either symmetry or the visual similarity across different access is not very salient for the model to capture it well, especially in the early stages of the training? - Some parts in the text could be written more clearly. For instance, -- could the authors explicitly explain what is a proper rotation matrix in line 97? -- what exactly is meant in l. 105-106 regarding solving the problem of the matrix being non positive semidefinite?

Correctness: Yes, the claims and the empircal methodology appear correct.

Clarity: Yes, the paper is generally clearly written.

Relation to Prior Work: Relation to prior work is well covered.

Reproducibility: Yes

Additional Feedback: Generally, I am positive of the submission, and I am lowering my score a bit because of the few weaknesses I listed above. I would be happy to raise my score with a convincing rebuttal.


Review 2

Summary and Contributions: This is a paper about modeling the distribution over SO(3) and using this distribution to compute the negative log-likelihood as a loss function in the training of a neural network, yielding a parameterized distribution as prediction. The authors provide an experimental study in 3D pose estimation from 2D images, rotated 3D model projections, and 3D poses from 2D head keypoints.

Strengths: This is a careful analysis of applying the matrix Fisher distribution as a representation for the conditional probability distribution output of a network learning an SO(3) element. A lot of calculations are taken from the excellent treatment in [11] and most of the theoretical grounding is in [11]. - The derivation of the constant of the Fisher distribution in the supplemental material is very helpful in obtaining an intuition about the influence of the singular values. - The authors show a method following the definition in [4] on how to compute the derivatives of the normalizing constant, a perplexing task given the combinatorial definition of the hypergeometric function. - Another strength is the convexity proof about the loss as well as the Lipschitz continuity of the loss itself and its derivatives. - An ablation analysis was run on the experiments with respect to data augmentation, class embedding, and homography preprocessing. - The method shows a superior performance in PAscal3D+ and on ModelNet10-SO(3).

Weaknesses: The following is a list of things I miss or I might have misunderstood and the authors should respond. - Regarding the backpropagation it is not obvious to the reader whether the pytorch SVD differentiation was used or any other method brewed by the authors (apologies for not looking at the provided code). - To understand the influence of the distribution I would appreciate an experiment when only the trace(F^T R) is used as a loss without a normalizing constant. A geodesic distance is used by Mahendran but to have a fair comparison it would be good to run the author's implementation just with the trace term. - While the authors provide qualitative results for ambiguous case with an elongated marginalized distribution ([11]-based visualization). However, they do not provide any table about using the 2nd moment of the contributions (formulae in [11]) and how it correlates to the actual accuracy (like figure 1 in [7]). That would be extremely helpful to see how well the Fisher distribution reflects accuracy. - The closest approach to this work is [5]. Unfortunately, they have not tried Modelnet10 or Pascal3D+ but the reader remains with the open question whether any probabilistic approach would perform better.

Correctness: The theory is correct and as a matter of fact the most interesting piece of the paper is the caluclations in the supplement. However, the empirical methodology lacks in two respects and leaves the reader with open questions: - Is the addition of the normalizing constant in the loss function the main factor that improved performance (vs just using the trace). - If this is indeed the case, authors have to compare with [5]. That would provide a validation for using the constant and would provide a straight comparison between Fisher and Bingham.

Clarity: The paper is very well written. Parts of the supplement on theory belong to the main paper (for sure the geometric interpretation).

Relation to Prior Work: - One more paper that containing useful formulae (preceding [11]) Sei, T., Shibata, H., Takemura, A., Ohara, K., & Takayama, N. (2013). Properties and applications of Fisher distribution on the rotation group. Journal of Multivariate Analysis, 116, 440-455. and a paper with a simple implementation Gaussian distributions on Lie groups and their application to statistical shape analysis PT Fletcher, S Joshi, C Lu, SM Pizer, 2003 might be worth citing. - The books by Greg Chirkjian. - Experimental comparison to [5].

Reproducibility: Yes

Additional Feedback:


Review 3

Summary and Contributions: The paper proposes using logarithm of the matrix Fisher distribution as a loss for training general purpose predictors of object orientation. To allow gradient based learning, the authors derive efficient method to compute gradient of the loss where the most important component is a hand-crafted approximation of the normalization function of the Fisher distribution. The method is empirically evaluated on three computer vision benchmarks and shown to achieve state-of-the-art results.

Strengths: Designing appropriate loss for learning object orientation predictors is a challenging problem with strong practical impact. The proposed model seems to be a valuable contribution which is both technically sound and provides empirical improvements against existing methods.

Weaknesses: I am missing an experiment that would compare the proposed loss with the existing alternatives in a controlled setting (i.e. using the same architecture, training algorithm, data and change only the loss) in order to clearly show see the differences/benefits. If I understand it correctly, the results reported for competing methods were adopted from corresponding papers using different settings.

Correctness: The paper does not contain explicit form of the use approximation of the normalization constant. Hence it is difficult to verify the main claim that the proposed loss is convex.

Clarity: yes

Relation to Prior Work: yes

Reproducibility: Yes

Additional Feedback: The learned model provides a posterior distribution over the rotation matrices. Therefore, instead of using the distribution mode as the prediction, one could in principle use the Bayes optimal plug-in rule $argmin_{\hat{R}} \int_{R}p(R|X) d(R,\hat{R})$. The question is how difficult it is to solve the inference problem. By the authors' own admission, a conceptual problem is unimodality of the distribution in presence of rotation symmetric objects. The paper may discuss more how to resolve the problem? The proposed loss has some properties (like convexity, unimodality etc) that are claimed to be important. It would be helpful to see which of these properties are missing/present in case of the competing methods. I have read the author feedback.

[Author Response · NeurIPS 2020]

We have read and appreciate all comments, due to page limit we will only address a subset of questions/concerns.

**R1: The approximation of the normalizing factor is inelegant** Since submitting the paper we have implemented a normalizing factor based on numerical integration of the equations (14) and (15) in [11]. The planned method uses 512 trapezoid integrals with float32 precision. We plan to use this new method for the camera ready, remove all mentions of the old method and rerun all experiments. We do not expect this will significantly change our performance.

**R1: Can you evaluate the coarseness of the approximation?** We use the integration method with $2^{14}$ trapezoids and float128 precision as the true function.

By sampling 1000 random singular values with an L2 norm $< 50$ we get the following results. The relative error of the forward pass $|log(\hat{a}(F)) - log(a(F))|/log(a(F))$ has an average error of 0.07 with the old method. This error for the integration method (512 trapezoids) is $\approx 10^{-5}$. The norm of error for the backward pass $||\nabla_F log(\hat{a}(F)) - \nabla_F log(a(F))||_F$ is for the old method on average $\approx 10^{-1}$. With the integration method (512 trapezoids) it is $\approx 10^{-3}$.

**R4: Approximation $\implies$ loss not necessarily convex**: This is true. Hopefully though as we are approximating a convex loss this still greatly helps with the convergence and stability of training. Also any standard approximation (look-up table, rbf as in [5], horner's method etc.) could potentially affect the convexity.

**R1: approximation $\implies$ estimated $R$ is not in $SO(3)$?** The approximation affects the predicted $F$ and thus the $R$ we estimate but not the property of it being in $SO(3)$. The estimated $R$, obtained from $F$ via the steps described in section 3.1, is guaranteed to be in $SO(3)$.

**R1: Do the model get stuck in bad local optima for classes with symmetry or visual similarity for different poses, especially early in the training** As described in line 219 - 229 What we do observe is that the network early on gives large variance for the axis which it cannot identify. This is not a local minima for the network outputs, since the optima is to predict the correct pose confidently and the loss is (approximately) convex. We observe a reasonably stable reduction in training loss (fig 2 in supplementary) this indicates that we do not get stuck in local minimas for long, if at all.

**R1: What do you mean by solving the problem of non-positive semidefinite matrices?** $_1F_1$ is defined for positive semidefinite matrices, see [2]. It can be extended to non positive semidefinite matrices as in appendix eq. (26) or in [2]. **Clarify "proper rotation matrix"?** A proper rotation matrix is ON and $det(R) = 1$

**R2: Is the posterior F well aligned with the empirical error?** We have not investigated this rigorously, but anecdotally this seems to be the case for large errors. For small errors it is probably not the case since we over regularize (line 115-117 in paper), this disincentives the network from returning low variance outputs.

**R2: Would using only the $tr(F^T R)$ as a loss work?** No. Without regularization it is possible to get arbitrarily low loss with high average angle errors by outputting large magnitude F's. Using other regularizers such as $||F||_F^2$ could work, but we have used a regularizer with a probabilistic motivation.

**R4: Could you do an ablation with more/all other competing losses?** Ideally yes, but we do think that the current ablation shows that even when removing data augmentations and our warp preprocessing we outperform other methods. We think this is strong evidence that our method is significantly better. Doing ablations can be problematic due to availability of code and sensitivity to tuning.

**R2: Do you use pytorch SVD for backpropagation?** Yes we mainly use pytorch svd. We do some custom handling to avoid instability from using $det(U^T V)$

**R2: Could you compare to [5]?** Empirically we evaluated on UPNA for this reason. (line 197-199) Their code was not available at the time of writing. Their loss is discontinous and non convex due to Gram-Schmidt.

**R2: Would any probabilistic approach perform better [than Current SOTA]** Not necessarily, [21] (probabilisitc) is outperformed by [16] (not probabilistic) which is outperformed by ours (table 1).

**R4: Instead of using the mode could you use an Bayes estimator for some distance?** Due to symmetry the two coincide for many reasonable distances. see Theorem 2.3 in [11] for one example.

**R1, R4: The method has a problem with rotation symmetric objects.** We developed this method with identifiable orientations in mind. We noticed some classes had rotation symmetries and we thought our method had interesting behaviour for these objects. Another distribution should be used when estimating orientation of symmetric objects.

**R4: The loss have the properties (Convexity, rotation invariance, bounded gradient etc.), which properties are present in competing methods?** We did not investigate this thoroughly. In general these properties are the exception not the rule. We believe methods which use quaternions(discontinuity), Euler angles(rotation invariance), classification(rotation invariance) or Gram-Schmidt (discontinuity) will lose at least one of these properties.

[Meta-Review · NeurIPS 2020]

The reviewers have generally supported this paper. I recommend the authors to address the reviewers' comments in the next revision.